# Increased Antiproliferative Activity of Antiestrogens and Neratinib Treatment by Calcitriol in HER2-Positive Breast Cancer Cells

**DOI:** 10.3390/ijms26178396

**Published:** 2025-08-29

**Authors:** Edgar Milo-Rocha, Lorenza Díaz, Janice García-Quiroz, Heriberto Prado-Garcia, Rocío García-Becerra

**Affiliations:** 1Departamento de Biología Molecular y Biotecnología, Instituto de Investigaciones Biomédicas, Universidad Nacional Autónoma de México, Ciudad de México 04510, Mexico; edgarmilorocha@iibiomedicas.unam.mx; 2Departamento de Biología de la Reproducción Dr. Carlos Gual Castro, Instituto Nacional de Ciencias Médicas y Nutrición Salvador Zubirán, Vasco de Quiroga No. 15, Belisario Domínguez Sección XVI, Tlalpan, Ciudad de México 14080, Mexico; lorenza.diazn@incmnsz.mx (L.D.); janice.garciaq@incmnsz.mx (J.G.-Q.); 3Laboratorio de Onco-Inmunobiología, Departamento de Enfermedades Crónico-Degenerativas, Instituto Nacional de Enfermedades Respiratorias Ismael Cosío Villegas, Calzada de Tlalpan 4502, Belisario Domínguez Sección XVI, Tlalpan, Ciudad de México 14080, Mexico; hpradog@yahoo.com; 4Programa de Investigación de Cáncer de Mama, Instituto de Investigaciones Biomédicas, Universidad Nacional Autónoma de México, Ciudad de México 04510, Mexico

**Keywords:** HER2-positive breast cancer, calcitriol, neratinib, tamoxifen, fulvestrant

## Abstract

HER2-positive breast cancer is an aggressive subtype, often associated with shorter progression-free and overall survival. Estrogen receptor (ER) expression within this subtype leads to distinct growth patterns and treatment responses. Calcitriol, the active form of vitamin D, induces ERα expression in ER-negative breast cancer cells, thereby sensitizing them to the antiproliferative effects of antiestrogens. When combined with neratinib, calcitriol enhanced cell growth inhibition. Therefore, we investigated whether adding calcitriol to the combined treatment with antiestrogens and neratinib could further inhibit the proliferation of HER2-positive breast cancer cells, regardless of their ER status. The BT-474 (ER-positive/HER2-positive) and SK-BR-3 (ER-negative/HER2-positive) breast cancer cell lines were pretreated with calcitriol to modulate ER expression, followed by treatment with calcitriol in combination with neratinib, with or without antiestrogens. Proliferation assays, cell cycle analysis, and Western blotting were then performed to assess treatment effects. The results demonstrated that calcitriol and neratinib, per se, significantly inhibited cell proliferation in a concentration-dependent manner in the HER2-positive cell lines. Notably, calcitriol enhanced the antiproliferative response of combined neratinib and antiestrogen treatment. Calcitriol, alone or in combination, modulated vitamin D receptor and ERα expression, reduced AKT and ERK phosphorylation, and promoted G1 phase arrest. These findings support the potential of this combinatorial approach as a therapeutic strategy for HER2-positive breast cancer.

## 1. Introduction

Breast cancer stands as the most diagnosed malignant tumor among women worldwide, and it ranks as the leading cause of cancer-related deaths. Over the past three decades, both its incidence and mortality rates have shown an alarming increase. Furthermore, breast cancer presents as a heterogeneous disease, exhibiting diverse biological characteristics and clinical outcomes [1].

Breast cancer has undergone extensive study, leading to its classification into distinct subtypes based on the expression of specific molecular markers. These subtypes are typically categorized into four groups: luminal A and B, HER2-enriched, and triple-negative breast cancer [2,3].

The luminal A and B subtypes are characterized by high expression of estrogen receptor alpha (ERα) and progesterone receptor (PR). Approximately 70% of all diagnosed tumors exhibit these characteristics, typically associated with a favorable prognosis. Patients with these phenotypes are suitable candidates for antiestrogenic therapy. Currently, three classes of antiestrogenic therapy are widely employed: aromatase inhibitors (AIs), such as anastrozole, letrozole, and exemestane; selective estrogen receptor modulators (SERMs), such as tamoxifen, which directly inhibit ER transcriptional activity by binding to it [4]; and selective estrogen receptor downregulators (SERDs) like fulvestrant, which promote destabilization and degradation of ER. These strategies have demonstrated clinical efficacy in women with ER-positive breast cancer [5,6,7,8].

Tumors that exhibit HER2 overexpression comprise approximately 15% of all breast cancer cases. They are characterized by the upregulation of genes associated with HER2 signaling and low expression levels of ERα and PR. These features have been correlated with an aggressive, metastatic phenotype and a poor prognosis [9,10].

Approximately 50% of HER2-positive breast cancers also express the ER, dividing patients into two main subgroups: ER-negative/HER2-positive and ER-positive/HER2-positive. These subtypes differ in their biological behavior and response to treatment. ER-negative tumors, which do not benefit from antiestrogen therapy due to the lack of ER expression, are typically treated with a combination of chemotherapy and anti-HER2 agents such as trastuzumab or neratinib, an irreversible pan-HER tyrosine kinase inhibitor [11,12].

Neratinib has been approved by the U.S. FDA for extended adjuvant treatment following trastuzumab-based therapy in early-stage HER2-positive breast cancer, and in combination with capecitabine for patients with advanced or metastatic HER2-positive disease who have received at least two prior anti-HER2 therapies [13,14]. Similarly, the European Medicines Agency approved its use in hormone receptor–positive, HER2-positive early breast cancer as extended adjuvant therapy [15].

In the ER-positive/HER2-positive breast cancer subgroup, several therapeutic strategies are currently under active investigation, including combinations such as aromatase inhibitors with lapatinib, an anti-HER2 tyrosine kinase inhibitor [16], and fulvestrant with trastuzumab [17]. These approaches have shown clinical benefits by incorporating antiestrogen therapy [18]. However, not all patients respond adequately, and resistance can develop over time. Therefore, identifying compounds that can enhance treatment efficacy and improve patient outcomes remains crucial.

In this regard, utilizing calcitriol, the most active metabolite of vitamin D, emerges as a promising strategy. Calcitriol not only demonstrates antineoplastic effects in cancer cells but also enhances the sensitivity of breast tumor cells to various chemotherapeutic agents, ionizing radiation, antiestrogenic compounds, and tyrosine kinase inhibitors, including neratinib [19,20,21]. The antineoplastic properties of calcitriol are mediated through several mechanisms via the vitamin D receptor (VDR), including growth inhibition, suppression of migration and invasion, and induction of cell differentiation and apoptosis [22,23,24,25]. Previous in vitro studies have demonstrated that calcitriol induces ERα expression in ER-negative breast cancer cells through both transcriptional activation and epigenetic mechanisms, sensitizing them to the antiproliferative effects of antiestrogens [26,27]. Moreover, combining calcitriol with neratinib more effectively inhibited cell proliferation and reduced AKT and MAPK phosphorylation compared to either compound alone [20]. Similarly, the calcitriol analog EB1089 enhanced the antiproliferative response of HER2-positive breast cancer cells when added to lapatinib–antiestrogen combinations, partly by regulating ERα protein expression and decreasing AKT phosphorylation [28].

Based on this evidence, we aimed to investigate the effect of adding calcitriol to the combined treatment with antiestrogens and neratinib in HER2-positive breast cancer. Specifically, we evaluated its impact on cell proliferation, VDR and ERα expression, AKT and ERK phosphorylation, and cell cycle distribution.

## 2. Results

### 2.1. Calcitriol and Neratinib Inhibit Proliferation of BT-474 and SK-BR-3 Breast Cancer Cells

The antiproliferative effects of calcitriol and neratinib were evaluated in two HER2-positive breast cancer cell lines with distinct ER status: BT-474 (ER-positive/HER2-positive) and SK-BR-3 (ER-negative/HER2-positive) [28].

Cells were treated for six days with increasing concentrations (1 × 10^−11^ M–1 × 10^−6^ M) of each compound, followed by proliferation assays. The results demonstrated that calcitriol inhibited cell proliferation in a concentration-dependent manner. However, the degree of sensitivity to compounds differed between the two cell lines (Figure 1A,B).

The inhibitory effect of calcitriol on the proliferation of BT-474 cells was significant from the concentration of 1 × 10^−9^ M compared to untreated cells (Figure 1A), whereas in SK-BR-3 cells, it was significant from the concentration of 5 × 10^−9^ M (Figure 1B).

Regarding the effect of the tyrosine kinase inhibitor on the proliferation of both cell lines, similar to the previous results, it was shown that neratinib inhibited cell proliferation in a concentration-dependent manner (Figure 1C,D). Specifically, the inhibitory effect was significant at a concentration of 1 × 10^−9^ M in BT-474 cells and 7.5 × 10^−10^ M in SK-BR-3 cells.

The IC_20_ and IC_50_ values were calculated from the concentration–response curves of each compound (Table 1). Considering the IC_50_ values, calcitriol obtained similar IC_50_ values in both cell lines, indicating comparable sensitivity. In contrast, BT-474 cells were more sensitive to neratinib than SK-BR-3 cells, with an approximately tenfold difference in IC_50_ values.

### 2.2. Calcitriol Enhances the Antiproliferative Effect of Antiestrogens and Neratinib in HER2-Positive Breast Cancer Cells

To investigate the potential of calcitriol as a sensitizing agent, we evaluated its effect in combination with neratinib and the antiestrogens tamoxifen and fulvestrant on the proliferation of HER2-positive breast cancer cells. For these experiments, BT-474 (ER-positive/HER2-positive) and SK-BR-3 (ER-negative/HER2-positive) cell lines were selected to represent distinct molecular subtypes that differ in their estrogen receptor status. Cells were treated using the previously determined IC_50_ values of calcitriol and neratinib (Table 1).

HER2-positive breast cancer cells were pretreated in the absence (V, black bars) or presence of calcitriol (C, white bars). Subsequently, cells were treated with calcitriol, neratinib (N, black bars with stripes), or a combination of both compounds (C + N, white bars with stripes). These treatments were carried out in the absence (Con) or presence of estradiol (E), tamoxifen (T), fulvestrant (F), or the combination of the antagonists with estradiol. Subsequently, proliferation assays were performed.

Figure 2A shows that estradiol per se and in combination with neratinib (E, black, and black bars with stripes) significantly increased BT-474 cell proliferation compared with untreated cells or those treated only with the tyrosine kinase inhibitor, respectively (Con, black, and black bars with stripes). In contrast, fulvestrant alone or combined with neratinib (F, black and black bars with stripes) significantly decreased cell growth relative to vehicle- or neratinib-treated controls (Con, black and black bars with stripes). Treatments with tamoxifen, either alone or in combination with neratinib (T, black, black bars with stripes), did not affect cell growth. Furthermore, the addition of antiestrogens to estradiol treatments (E + T and E + F, black bars) or to estradiol plus neratinib (E + T and E + F, black bars with stripes) significantly reversed the proliferative effect induced by estradiol or estradiol with neratinib, respectively (E, black and black bars with stripes). The observed sensitivity to fulvestrant was expected, given that BT-474 cells express ERα and therefore respond to this treatment.

Cells treated with calcitriol alone or in combination with neratinib (Con, white, and white bars with stripes) showed a significant reduction in cell proliferation compared with untreated cells or those treated only with the tyrosine kinase inhibitor (Con, black, and black bars with stripes).

Remarkably, the addition of calcitriol, either alone or combined with neratinib, to treatments with estradiol, antiestrogens, or their combination (E, E + T, E + F, T, F, white bars or white bars with stripes) further decreased cell growth compared with the corresponding groups without calcitriol or with the tyrosine kinase inhibitor (E, E + T, E + F, T, F, black, and black bars with stripes). Notably, the triple combination of calcitriol, fulvestrant, and neratinib (F, white bar with stripes) exerted a significantly stronger inhibitory effect on cell proliferation than any of the individual treatments (Con, white bar; F, black bar, Con, black bar with stripes), as well as the dual combinations of fulvestrant plus neratinib (F, black bar with stripes) or calcitriol plus neratinib (Con, white bar with stripes).

These results indicate that calcitriol, alone or in combination with neratinib, enhances the antiproliferative effects of antiestrogens in ER-positive/HER2-positive breast cancer cells.

The absence of ERα in breast cancer is associated with poor prognosis, increased risk of migration, metastasis, and resistance to endocrine treatment. Therefore, we evaluated the effect of adding calcitriol to antiestrogens and neratinib treatment in the SK-BR-3 breast cancer cell line. As shown in Figure 2B, in the absence of calcitriol (black bars), estradiol alone (E), estradiol combined with antiestrogens (E + T, E + F), or tamoxifen (T) and fulvestrant (F), did not modify cell proliferation. These results were expected due to the lack of ERα expression.

The treatments of calcitriol, neratinib, and their combination (Con and E, white, black with stripes, and white bars with stripes) inhibited cell proliferation, both in the absence or presence of estradiol (Con and E, black bars). This inhibitory effect was also observed when estradiol was combined with antiestrogens or when antiestrogens were administered alone (E + T, E + F, T and F, white, black with stripes, and white bars with stripes). In all cases, proliferation was significantly reduced compared to untreated cells (Con, black bar) and to the corresponding groups without calcitriol (E, E + T, E + F, T, F, black bars).

The addition of calcitriol to neratinib significantly enhanced the inhibition of cell proliferation compared with calcitriol alone in all groups (E, E + T, E + F, F, white bars with stripes) except in the tamoxifen group (T, white bar with stripes). Similarly, combining calcitriol with neratinib (E and F, white bars with stripes) produced a greater inhibitory effect than neratinib alone in the estradiol and fulvestrant groups (E and F, black bars with stripes). Notably, in the latter, the triple treatment calcitriol, fulvestrant, and neratinib (F, white bar with stripes) exerted the strongest antiproliferative effect, showing a statistically significant difference compared with each treatment administered separately (Con, white; F, black; Con, black with stripes; F, white; Con, white with stripes; F, black with stripes).

These results indicate that calcitriol enhances the antiproliferative effects of fulvestrant and neratinib in ER-negative/HER2-positive breast cancer cells.

Based on these data, we decided to analyze the expression of key proteins that play a fundamental role in proliferation, invasion, and metastasis, which are principal characteristics of breast cancer.

Given tamoxifen’s widespread clinical use and well-established efficacy [29,30], we focused the subsequent mechanistic analyses on this antiestrogen to investigate its combined effects with calcitriol and neratinib on key proteins involved in proliferation, invasion, and metastasis.

### 2.3. Calcitriol Upregulates VDR Expression in HER2-Positive Breast Cancer Cell Lines

To determine the effect of treatments on VDR expression, we performed Western blot analysis in BT-474 and SK-BR-3 cells pretreated in the absence or presence of calcitriol and subsequently treated with tamoxifen (Tx), neratinib (Ner), or their combination (Figure 3).

In BT-474 (Figure 3A) and SK-BR-3 (Figure 3B) cells, treatment with calcitriol alone increased VDR protein levels compared to vehicle-treated controls. This upregulation of VDR was maintained when calcitriol was combined with tamoxifen (Cal + Tx), neratinib (Cal + Ner), or the triple combination (Cal + Tx + Ner). Notably, the latter regimen induced the highest increase in VDR expression, as confirmed by densitometric analysis normalized to β-actin.

These results suggest that calcitriol-mediated VDR upregulation may enhance the sensitivity of HER2-positive breast cancer cells to antineoplastic treatments.

### 2.4. Differential Regulation of ERα by Calcitriol in HER2-Positive Breast Cancer Cells Treated with Tamoxifen and Neratinib

In BT-474 cells (Figure 4A), ERα expression was detectable under control (Vh) conditions, as expected; all treatments led to a reduction in ERα levels compared with vehicle-treated cells. Notably, the combination of calcitriol, tamoxifen, and neratinib (Cal + Tx + N) produced the most pronounced decrease in ERα expression, as confirmed by densitometric analysis normalized to β-actin.

In contrast, ERα expression was not detectable under control (Vh) conditions in SK-BR-3 cells (Figure 4B), consistent with their ER-negative status. Interestingly, treatment with calcitriol (Cal) alone induced ERα expression, and in all conditions that included calcitriol (Cal + Tx, Cal + Ner, and Cal + Tx + Ner), ERα expression was consistently observed.

These results are particularly relevant because they corroborate that calcitriol can induce ERα in ER-negative cells, thereby restoring the molecular target required for antiestrogen therapies.

### 2.5. Combined Treatment with Calcitriol, Tamoxifen, and Neratinib Inhibits AKT Activation in HER2-Positive Breast Cancer Cells

In BT-474 cells (Figure 5A), treatment with calcitriol either alone (Cal) or in combination with tamoxifen (Cal + Tx), neratinib (Cal + Ner), or both (Cal + Tx + Ner) reduced AKT phosphorylation compared to vehicle-treated control. A similar decrease was also observed with the tamoxifen (Tx) alone, as confirmed by densitometric analysis normalized to total AKT.

In SK-BR-3 cells (Figure 5B), most treatments with or without calcitriol (Tx, Ner, Tx + Ner, Cal, Cal + Tx, and Cal + Tx + Ner) decreased pAKT levels compared to untreated cells (Vh), except in the calcitriol and neratinib (Cal + Ner) combination, which did not reduce pAKT. The most pronounced inhibition of AKT phosphorylation was observed with the triple combination (Cal + Tx + Ner).

Together, these results indicate that calcitriol suppresses AKT signaling in HER2-positive breast cancer cells, thereby potentially enhancing the antiproliferative efficacy of antiestrogens and tyrosine kinase inhibitors.

### 2.6. Combined Treatment with Calcitriol, Tamoxifen, and Neratinib Inhibits ERK Activation in HER2-Positive Breast Cancer Cells

In BT-474 cells (Figure 6A), treatments in the absence of calcitriol, including tamoxifen (Tx), neratinib (Ner), and their combination (Tx + Ner), as well as calcitriol (Cal) alone, did not modify pERK levels compared to vehicle-treated controls. In contrast, a pronounced decrease in ERK phosphorylation was observed when calcitriol was administered in combination with either tamoxifen (Cal + Tx), neratinib (Cal + Ner), or both (Cal + Tx + Ner), as confirmed by densitometric analysis normalized to total ERK.

In SK-BR-3 cells (Figure 6B), a similar trend was observed, with the greatest reduction in ERK phosphorylation occurring when calcitriol was combined with tamoxifen (Cal + Tx), neratinib (Cal + Ner), or both (Cal + Tx + Ner).

These findings indicate that calcitriol effectively suppresses ERK activation, particularly when used in combination with antiestrogens and tyrosine kinase inhibitors, supporting its potential role in modulating MAPK/ERK signaling in HER2-positive breast cancer cells.

### 2.7. Triple Combination of Calcitriol, Tamoxifen, and Neratinib Induces G1 Arrest in HER2-Positive Breast Cancer Cells

Cell cycle analysis (Figure 7) showed that addition of calcitriol to tamoxifen (C + T, light gray bar with stripes) or to the combination of tamoxifen and neratinib (C + T + N, black bar with stripes) significantly increased the proportion of BT-474 cells in the G1 phase compared with the vehicle (white bar with stripes). This effect was also significant when compared with tamoxifen or neratinib alone (T and N, light and dark gray bars, respectively). Moreover, calcitriol plus tamoxifen (C + T, light gray bar with stripes) further increased G1 arrest compared with tamoxifen alone (light gray bar). No significant differences were observed in the Sub G0, S, or G2/M phases across treatments.

These results indicate that calcitriol, in combination with tamoxifen and neratinib, promotes G1 arrest in BT-474 cells.

## 3. Discussion

Despite the diversity of pharmacological strategies currently available for the treatment of breast cancer, therapeutic resistance remains a major challenge that significantly compromises clinical outcomes and patient quality of life. One of the critical factors contributing to this resistance is the activation of compensatory signaling pathways, which result from the intricate bidirectional crosstalk between hormone receptors and growth factor receptors within tumor cells. This molecular plasticity limits the long-term efficacy of targeted monotherapies, facilitating tumor progression and metastasis [31].

Neratinib, an irreversible pan-HER tyrosine kinase inhibitor, has demonstrated considerable clinical efficacy, particularly in HER2-positive breast cancer. However, its use as monotherapy is often limited by the development of acquired resistance over time [32,33]. To overcome this limitation, combination strategies that concurrently target multiple oncogenic pathways have gained attention. In particular, the dual inhibition of HER2 and ER signaling has shown promise in ER-positive/HER2-positive tumors, a biologically distinct and clinically relevant breast cancer subtype [34].

Antiestrogens, such as tamoxifen and fulvestrant, inhibit estrogen signaling by blocking ER activation, thereby suppressing tumor cell proliferation. Tamoxifen is the most commonly used endocrine therapy for ER-positive breast cancer, employed in both early and advanced disease, and as a preventive agent in high-risk women [35,36]. Fulvestrant is effective and well-tolerated, particularly in postmenopausal patients with advanced disease who are naïve to or progressing on prior hormonal therapy [37,38]. However, patients with ERα-negative tumors derive little benefit from antiestrogen therapies due to the lack of receptor expression [39,40], underscoring the need for alternative or sensitizing strategies in this subgroup.

In this context, calcitriol, the hormonally active form of vitamin D, has emerged as a promising therapeutic agent due to its well-documented antiproliferative, pro-differentiation, and pro-apoptotic effects in various cancer models, including breast cancer [41]. Previous studies have demonstrated that the addition of calcitriol can enhance the efficacy of conventional therapies, including chemotherapeutic agents and targeted treatments [20,24]. These biological properties support the rationale for evaluating calcitriol as a component of combination therapies aimed at overcoming resistance mechanisms in HER2-positive breast cancer. This includes not only tumors with ER positivity but also those lacking ER expression or exhibiting resistance to endocrine therapy.

Our study demonstrates that both calcitriol and neratinib exert concentration-dependent antiproliferative effects in HER2-positive breast cancer cell lines. Calcitriol exhibited similar IC_50_ values in BT-474 and SK-BR-3 cells, indicating comparable sensitivity, while BT-474 cells were approximately ten times more sensitive to neratinib than SK-BR-3 cells. These differences may be attributed to intrinsic molecular characteristics of each cell line, such as ER expression, HER2 status, and tumor aggressiveness, which influence therapeutic response.

In ER-positive/HER2-positive breast cancer cells, the co-expression of ER and HER2 activates a complex network of signaling pathways characterized by bidirectional crosstalk, ultimately promoting tumor aggressiveness and therapeutic resistance. Consequently, recent therapeutic strategies have focused on the use of combination therapies that target multiple oncogenic pathways simultaneously to enhance efficacy. Such combination approaches aim to inhibit tumor growth and metastasis, arrest cell proliferation, reduce cancer stem cell populations, and induce apoptosis [42].

Consistent with this rationale, our results show that the addition of calcitriol to the combined treatment with neratinib and antiestrogens (tamoxifen or fulvestrant) significantly enhances the antiproliferative response in both HER2-positive cell lines. These findings are consistent with previous studies, which report that calcitriol enhances the action of antineoplastic agents by modulating cell cycle progression, inducing apoptosis, and inhibiting proliferative signaling [20,24,41].

Specifically, in BT-474 cells, which co-express ERα and HER2, treatment with estradiol alone or in combination with neratinib significantly increased cell proliferation, underscoring the proliferative role of ER signaling in this subtype. In contrast, fulvestrant either alone or combined with neratinib markedly reduced proliferation, while tamoxifen had no significant effect, consistent with its partial agonist activity in certain cellular contexts [6,43]. Calcitriol, administered alone or in combination with neratinib, significantly inhibited proliferation, and its addition to endocrine therapies further enhanced the antiproliferative response, regardless of the presence of estradiol. Notably, the triple combination of calcitriol, neratinib, and fulvestrant produced the strongest inhibitory effect on cell growth, suggesting a synergistic effect that counteracts estradiol-induced proliferation.

Importantly, these findings demonstrate that the antiproliferative effects of calcitriol can be maintained even in the presence of estradiol and may even enhance the antineoplastic effect of tamoxifen, either alone or in combination with estradiol. These results are consistent with previous studies, which have shown the importance of calcitriol in enhancing the inhibitory effects of tamoxifen in both ER-negative and HER2-positive breast cancer cells [27,28].

In SK-BR-3 cells, which lack ERα expression, treatment with estradiol, antiestrogens, or their combinations did not significantly affect cell proliferation, as expected. In contrast, calcitriol and neratinib, either alone or in combination, effectively inhibited cell growth. The strongest antiproliferative effect was observed with the triple combination of calcitriol, neratinib, and fulvestrant. These results highlight the therapeutic potential of this strategy even in ER-negative/HER2-positive contexts and underscore the need for future in vivo studies in HER2-positive breast cancer models with differential ER expression. Such studies will be essential to evaluate the efficacy and translational relevance of triple therapies involving calcitriol, neratinib, and antiestrogens, particularly fulvestrant, in clinically relevant settings.

The re-sensitization to endocrine therapy and the enhanced antiproliferative response observed in SK-BR-3 cells may be attributed to the ability of calcitriol to restore ERα expression, as previously demonstrated in our laboratory [26]. This sensitizing effect is likely VDR-dependent, since in a previous study we showed that calcitriol increased ERα mRNA in a dose-dependent manner, an effect abolished by the VDR antagonist TEI-9647 [26]. Additionally, calcitriol exerts intrinsic antiproliferative effects that improve responses to conventional therapies used in breast cancer, such as neratinib and antiestrogens. In this context, several studies have shown that calcitriol enhances the antiproliferative efficacy of various antineoplastic agents when used in combination [19,20,28].

Collectively, these findings support the use of calcitriol as an adjuvant to improve the therapeutic efficacy of neratinib and antiestrogens in both ERα-positive and ERα-negative HER2-positive breast cancer.

Due to its widespread clinical use and robust evidence base, including randomized trials demonstrating that five years of tamoxifen treatment reduces breast cancer recurrence by approximately 40% and mortality by one-third, tamoxifen remains a cornerstone of adjuvant endocrine therapy in hormone receptor-positive breast cancer, particularly among premenopausal women [29,30]. Therefore, we focused on tamoxifen in our mechanistic studies to evaluate its effects in combination with calcitriol and neratinib on the expression of key proteins involved in proliferation, invasion, and metastasis. This approach strengthens the translational relevance of our findings by aligning them with current therapeutic strategies.

Calcitriol exerts its biological effects primarily through binding to VDR, a nuclear transcription factor expressed in various tissues, including breast tissue. Upon activation, VDR regulates multiple cellular processes, including calcium transport, signal transduction, cell proliferation, and differentiation [44,45]. In addition, VDR expression levels may correlate with the cells’ responsiveness to calcitriol. In this study, we demonstrate that calcitriol upregulates VDR expression in HER2-positive breast cancer cells, regardless of their ER status. Moreover, the combination of calcitriol with antiestrogens and neratinib further enhances VDR levels, particularly under triple treatment conditions. This upregulation may contribute to the increased sensitivity of these cells to antineoplastic therapies, supporting the potential role of calcitriol as an adjuvant agent in the treatment of HER2-positive breast cancer. Notably, the upregulation of VDR by calcitriol observed in this study aligns with previous findings reported by García-Quiroz et al., who found that calcitriol treatment led to increased VDR expression in breast cancer cells [46]. Understanding VDR expression in these cell lines could provide insights into their differential responses to calcitriol-based therapies.

In breast cancer, the presence of ER is generally considered a favorable prognostic marker, as it is associated with less aggressive tumor biology, longer disease-free intervals, and improved overall survival compared to ER-negative tumors [47]. Patients with ER-negative/HER2-positive tumors tend to experience earlier relapses and a more aggressive clinical course than those with ER-positive/HER2-positive disease [48]. Moreover, ER-positive tumors are more likely to respond to endocrine therapies, reinforcing the therapeutic value of ER expression [49,50,51].

In our study, the addition of calcitriol to tamoxifen and neratinib downregulated ERα protein levels in BT-474 cells (ER-positive/HER2-positive) and induced its expression in SK-BR-3 cells (ER-negative/HER2-positive). These results are consistent with previous reports showing that calcitriol modulates ERα expression depending on the initial receptor status [26,27,52]. In luminal cells, such as BT-474, the downregulation of ERα may be partially mediated by the binding of the calcitriol–VDR-RXR complex to negative vitamin D response elements (nVDREs) within the ESR1 promoter, leading to transcriptional repression [53].

Conversely, in SK-BR-3 cells, calcitriol alone or in combination induces ERα expression, corroborating previous findings by Santos-Martínez et al. [26], who demonstrated re-expression of ERα in ER-negative breast cancer cells following calcitriol treatment. This effect has been attributed to both direct transcriptional regulation and epigenetic modifications [27]. Notably, treatment with tamoxifen and its combination with neratinib also promoted ERα expression in SK-BR-3 cells, which may result from the inhibition of HER2-driven signaling and the activation of compensatory ERα pathways, highlighting the complex crosstalk between these signaling axes [54]. These results are consistent with previous observations from our laboratory, where we found that treatment with antiestrogens and the tyrosine kinase inhibitor lapatinib led to the upregulation of ERα in ER-negative/HER2-positive breast cancer cells [28].

Together, these findings indicate that calcitriol modulates ERα expression in a receptor status-dependent manner, potentially reversing features associated with endocrine resistance and aggressive tumor behavior. This context-dependent modulation underscores the potential of calcitriol as a therapeutic adjuvant to improve the efficacy of combined treatment strategies in HER2-positive breast cancer.

In breast cancer, particularly in the HER2-positive subtype, aberrant activation of the PI3K/AKT/mTOR and mitogen-activated protein kinase (MAPK/ERK) signaling pathways is frequently observed, promoting tumor cell proliferation, growth, migration, invasion, survival, and therapeutic resistance [55,56,57,58,59,60]. The PI3K/AKT/mTOR pathway regulates critical cellular processes, including growth, metabolism, and the inhibition of apoptosis, with AKT phosphorylation (pAKT) serving as a key marker of its activation. Similarly, the MAPK/ERK pathway, a highly interconnected cascade involving various kinases, plays a central role in oncogenesis and the development of drug resistance. Due to their relevance in breast cancer progression, these pathways have become major targets for therapeutic intervention [55,56,57,58,59,60,61]. In this context, we evaluated the effect of combining calcitriol with neratinib and tamoxifen on the phosphorylation of AKT and ERK in HER2-positive breast cancer cells. Our results demonstrate that treatment with calcitriol, either alone or in combination with tamoxifen and neratinib, reduced phosphorylation levels of both AKT and ERK in HER2-positive breast cancer cells. This suggests that calcitriol can disrupt key survival and proliferative signaling mechanisms, potentially enhancing the efficacy of antiestrogens and tyrosine kinase inhibitors.

These findings are consistent with previous reports by Segovia-Mendoza et al. [20], who showed that the combination of calcitriol and neratinib decreased pAKT and pERK levels in triple-negative breast cancer cells. Taken together, our results support the notion that calcitriol suppresses both PI3K/AKT and MAPK/ERK pathways in HER2-positive breast cancer cells, thereby reinforcing its potential as an adjuvant agent in combinatorial therapeutic strategies aimed at improving treatment outcomes and overcoming resistance. Considering these results, future studies evaluating the effects of calcitriol-based combinations on cell migration and invasion will be of great importance to further elucidate its role in limiting breast cancer progression.

Several studies have confirmed the antiproliferative effects of calcitriol in a wide range of normal and cancerous cells. Calcitriol regulates various cell cycle factors, either directly or through vitamin D response elements (VDREs) in their promoter regions. In breast cancer cells, at least three functional VDREs have been identified in the promoters of the cyclin-dependent kinase inhibitors p21 and p27. These early transcriptional events are associated with G0/G1 cell cycle arrest and reduced proliferation. Similar mechanisms have been described in other cancer models, including prostate, mammary, and mesenchymal cells. Notably, calcitriol may exert synergistic effects when combined with clinically used antineoplastic agents. Based on this evidence, we investigated the impact of combining calcitriol with tamoxifen and neratinib on cell cycle progression [62,63].

The cell cycle analysis revealed that calcitriol, either alone or in combination with other agents, predominantly induces cell cycle arrest at the G1 phase, as evidenced by a marked increase in the G1 population and a concomitant decrease in the S phase. Treatments with tamoxifen or neratinib alone resulted in an elevated sub-G0 population, suggesting the induction of apoptosis. However, the combination of calcitriol with tamoxifen and/or neratinib reduced the Sub G0 fraction, indicating a shift from a cytotoxic to a cytostatic response. The most pronounced effect was observed with the triple combination, which resulted in a significant accumulation of cells in the G1 phase and a reduction in the S phase, highlighting a strong antiproliferative effect. These results suggest that the combination of calcitriol with tamoxifen and neratinib enhances the antiproliferative response primarily by promoting G1 cell cycle arrest and reducing S-phase progression in BT-474 cells. Notably, the triple treatment exerted the most pronounced effect, indicating that calcitriol enhances the efficacy of the combination therapy by modulating cell cycle dynamics rather than inducing cell death.

In summary, our findings provide compelling evidence that calcitriol enhances the antiproliferative effects of antiestrogens and the HER2 inhibitor neratinib in HER2-positive breast cancer cells, regardless of their ER status. Calcitriol not only modulated the expression of ERα and VDR but also significantly reduced the activation of key oncogenic pathways, including the AKT and ERK signaling pathways. Additionally, it induced G1 cell cycle arrest, favoring a cytostatic rather than cytotoxic cellular response when combined with tamoxifen and neratinib. These pleiotropic effects highlight calcitriol’s potential as a therapeutic adjuvant to overcome resistance mechanisms and improve the efficacy of existing targeted therapies in HER2-positive breast cancer. Future studies, including in vivo models and clinical investigations, are warranted to validate these results and explore the translational applicability of calcitriol-based combination therapies in the clinical management of breast cancer.

## 4. Materials and Methods

### 4.1. Reagents

Cell culture media were obtained from Life Technologies (Grand Island, NY, USA). Fetal bovine serum (FBS) was purchased from Hyclone Laboratories Inc. (Logan, UT, USA). Neratinib was sourced from Sequoia Research Products (Pangbourne, UK). Calcitriol (1α, 25-dihydroxyvitamin D3), estradiol (E2), and 4-hydroxytamoxifen were purchased from Sigma (St. Louis, MO, USA). The antiestrogen fulvestrant (ICI 182,780) was obtained from Zeneca Pharmaceuticals (Wilmington, DE, USA).

### 4.2. Cell Line Culture

The HER2-positive cell lines SK-BR-3 (ER-negative/HER2-positive) and BT-474 (ER-positive/HER2-positive) were cultured according to the supplier’s instructions. The cell lines were obtained from the American Type Culture Collection (ATCC; Rockville, MD, USA). The cell lines were cultured in Hybrid-Care medium for BT-474 and McCoy’s 5A medium for SK-BR-3, both supplemented with inactivated 10% fetal bovine serum (FBS) and 1% penicillin (100 U/mL)/streptomycin (100 μg/mL). All experimental procedures were conducted in Dulbecco’s modified Eagle’s medium (DMEM) supplemented with 10% charcoal-stripped, heat-inactivated FBS, 100 U/mL penicillin, and 100 µg/mL streptomycin.

### 4.3. Treatments

BT-474 or SK-BR-3 cells were treated with neratinib or calcitriol in increasing concentrations (1 × 10^−11^ M to 1 × 10^−6^ M) for six days. Additionally, the cells were pretreated with calcitriol (IC_50_) or its vehicle (0.01% ethanol) as a control for 48 h to modulate ERα expression. Subsequently, the cells were treated with ethanol (V), IC_50_ of neratinib (N) BT-474 and SK-BR-3, respectively, estradiol (E, 1 × 10^−8^ M), tamoxifen (T, 1 × 10^−6^ M), fulvestrant (F, 1 × 10^−6^ M), or combinations of antagonists with estradiol. These treatments were carried out in the absence or presence of calcitriol. To assess the mechanistic effects on protein expression, the cells were pretreated with calcitriol and subsequently treated with tamoxifen, neratinib, or their combinations. The treatment durations for protein level and cell proliferation analyses were 48 h and six days, respectively.

### 4.4. Cell Proliferation Assay

SK-BR-3 and BT-474 cells were seeded in Corning^®^ 96-well plates at a specific density determined by a previously established growth curve. Cell proliferation was evaluated using the sulforhodamine B (SRB) colorimetric assay, as previously described [20]. Briefly, the cells were fixed with 10% trichloroacetic acid and incubated at 4 °C for 1 h. The plates were then gently washed with tap water and air-dried. Next, the cells were stained with 0.4% SRB dissolved in 1% acetic acid for 1 h at room temperature. To remove the unbound stain, the plates were washed four times with 1% acetic acid and air-dried again. The bound protein stain was then solubilized with 10 mM unbuffered Tris base [tris(hydroxymethyl) aminomethane]. Absorbance was measured at 492 nm using a microplate reader (Synergy HT Multi-Mode Microplate Reader, BioTek, Winooski, VT, USA). The IC_20_ and IC_50_ values of calcitriol were determined from the dose–response curve using the dose–response fitting function of the scientific graphing and analysis software OriginPro 8 (OriginLab Corporation, Northampton, MA, USA, version 8.0).

### 4.5. Western Blot Analysis

After treatments, whole-cell protein lysates were prepared using lysis buffer (50 mM Tris-HCl, 150 mM NaCl, 1% Nonidet P-40, pH 7.5) supplemented with a protease inhibitor cocktail. Protein concentrations were determined using the Protein Assay Dye Reagent Concentrate (Bio-Rad, Hercules, CA, USA). The proteins were separated on 10% SDS-PAGE and transferred to nitrocellulose membranes. The membranes were then blocked with 5% skim milk and incubated overnight at 4 °C with mouse anti-ERα antibody (1:500, Santa Cruz Biotechnology, Santa Cruz, CA, USA), anti-pAKT, anti-AKT (1:1000 Cell Signaling Technology, Danvers, MA, USA), anti-phospho-p44/42 MAPK ERK1/2, and anti-p44/42 MAPK ERK1/2 (1:1000 Cell Signaling Technology, MA, USA). After washing, the membranes were incubated with goat anti-mouse HRP-conjugated secondary antibody (1:2000, Santa Cruz Biotechnology, CA, USA). Membranes were visualized using BM chemiluminescence blotting substrate (Roche Applied Science, Mannheim, Germany). For normalization, blots were stripped in boiling stripping buffer (2% *w*/*v* SDS, 62.5 mM Tris-HCl, pH 6.8, 100 mM 2-mercaptoethanol) for 30 min at 50 °C and sequentially incubated with mouse anti-β-actin (1:1000, Santa Cruz Biotechnology, CA, USA) and anti-mouse-HRP (1:1000, Jackson ImmunoResearch Laboratories, Inc., West Grove, PA, USA).

### 4.6. Cell Cycle Distribution

Cells were incubated for 48 h with IC_50_ concentrations of neratinib and calcitriol, treated individually or in combination with tamoxifen. Following treatment, the cells were trypsinized and washed with phosphate-buffered saline (PBS, pH 7.4), fixed in 70% (*v*/*v*) ethanol, and stored at –20 °C. Prior to analysis, the samples were washed with PBS and incubated in the dark for 30 min in a solution containing PBS, Triton X-100, and propidium iodide (1 µg/mL). The cell cycle phases of each sample were analyzed using a BD Accuri™ C6 Plus Personal Flow cytometer, with a total of 10,000 events acquired from the PI-area versus PI-wide gate. The results were subsequently evaluated using FlowJo Software V10 (Beckton Dickinson, San Diego, CA, USA).

### 4.7. Statistical Analysis

Data are expressed as the mean ±standard deviation. Statistical differences were determined by one-way ANOVA followed by the Holm–Sidak test using specialized software (SigmaStat, Jandel Scientific version 3.5). Values of *p* < 0.05 or 0.001 were considered statistically significant. Prior to applying ANOVA, we assessed the normality of each dataset using the Kolmogorov–Smirnov test. The results showed that data from the SK-BR-3 cell line groups conformed to a normal distribution, while those from the BT-474 cell line groups did not. Accordingly, we applied parametric ANOVA to the SK-BR-3-derived data and non-parametric Kruskal–Wallis one-way analysis of variance on rank tests to the BT-474-derived data.

## 5. Conclusions

Calcitriol and neratinib individually exert significant antiproliferative effects in HER2-positive breast cancer cell lines in a concentration-dependent manner. Notably, the addition of calcitriol to the combined treatment with antiestrogens and neratinib further enhanced the antiproliferative response. This effect was associated with the modulation of VDR and ERα expression, the inhibition of AKT and ERK phosphorylation, and the induction of G1 phase arrest. Together, these findings support the use of calcitriol as a promising adjuvant agent to enhance the efficacy of current therapeutic regimens in HER2-positive breast cancer, particularly in tumors with low or absent ER expression or reduced responsiveness to endocrine therapy.

## Figures and Tables

**Figure 1 ijms-26-08396-f001:**
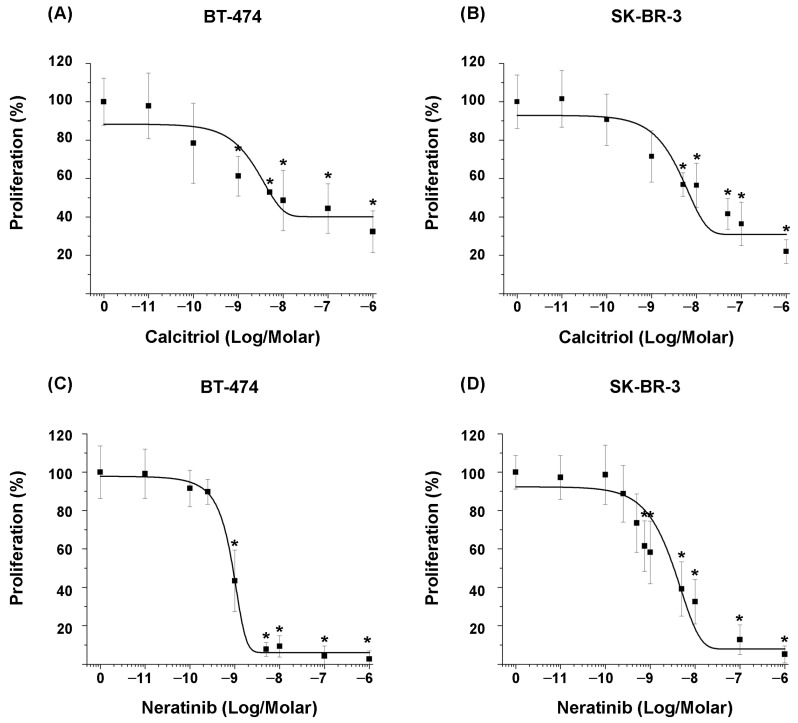
Antiproliferative effect of calcitriol and neratinib on BT-474 (ER-positive/HER2-positive) and SK-BR-3 (ER-negative/HER2-positive) breast cancer cells. Cells were treated for six days with increasing concentrations of (**A**,**B**) calcitriol or (**C**,**D**) neratinib. Cell proliferation was assessed using the sulforhodamine B colorimetric assay. Results are expressed as mean ± SD of three independent experiments. Data from vehicle-treated cells (0) were normalized to 100%. * *p* ≤ 0.001 vs. vehicle.

**Figure 2 ijms-26-08396-f002:**
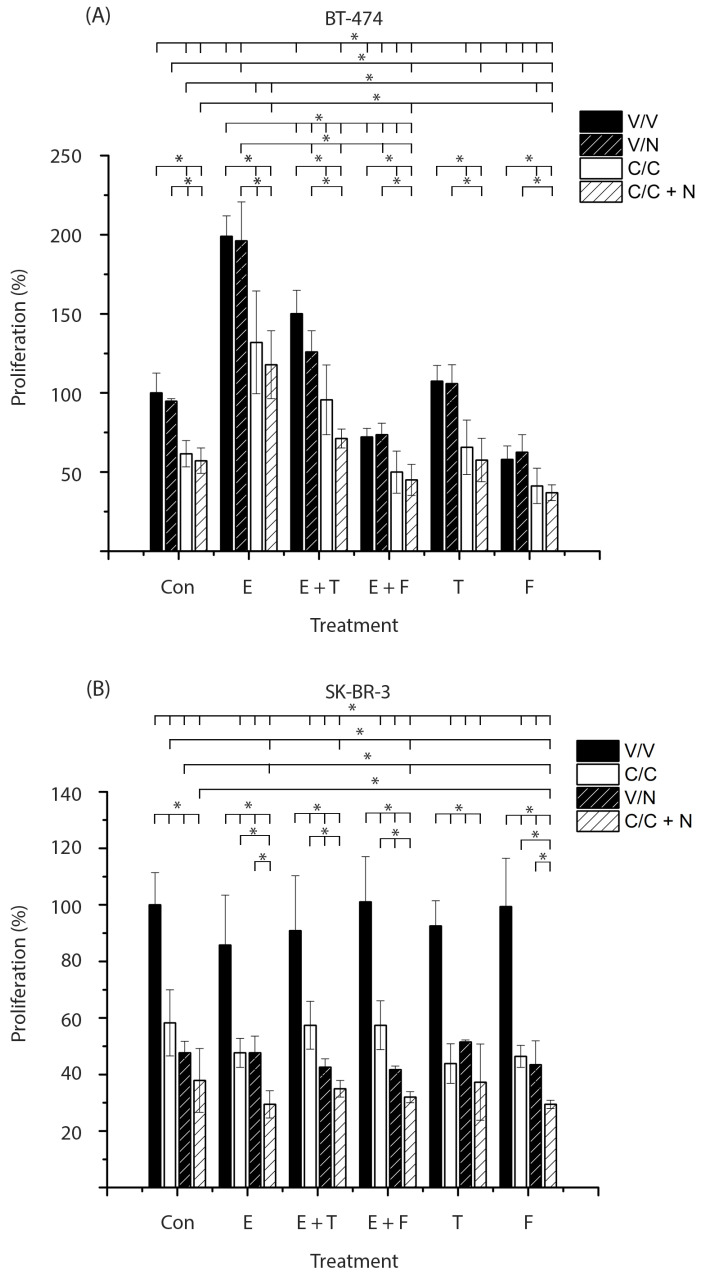
Effect of calcitriol on the proliferative response to neratinib and antiestrogens in HER2-positive breast cancer cells: (**A**) BT-474 and (**B**) SK-BR-3 cells were pretreated for 48 h with vehicle (V, black bars) or calcitriol (C, white bars; IC_50_ corresponding to each cell line). After pretreatment, cells were treated with neratinib (N; IC_20_ for BT-474 and IC_50_ for SK-BR-3; black bars with stripes), calcitriol (C, white bars), or the combination of both compounds (C + N, white bars with stripes). Treatments were carried out in the absence (Con) or presence of estradiol (E; 1 × 10^−8^ M), tamoxifen (T; 1 × 10^−6^ M), fulvestrant (F; 1 × 10^−6^ M), or the combination of estradiol with either antagonist. Bars represent mean ± SD from three independent experiments. Data were normalized to 100% relative to vehicle-treated controls. Statistical comparisons were performed as follows: V vs. all treatments; N, C, and C + N vs. groups containing N, C, and C + N, respectively; and in BT-474 cells, E and E + N vs. groups containing E and E + N, respectively. Within-group comparisons between treatments were also performed. * *p* ≤ 0.05.

**Figure 3 ijms-26-08396-f003:**
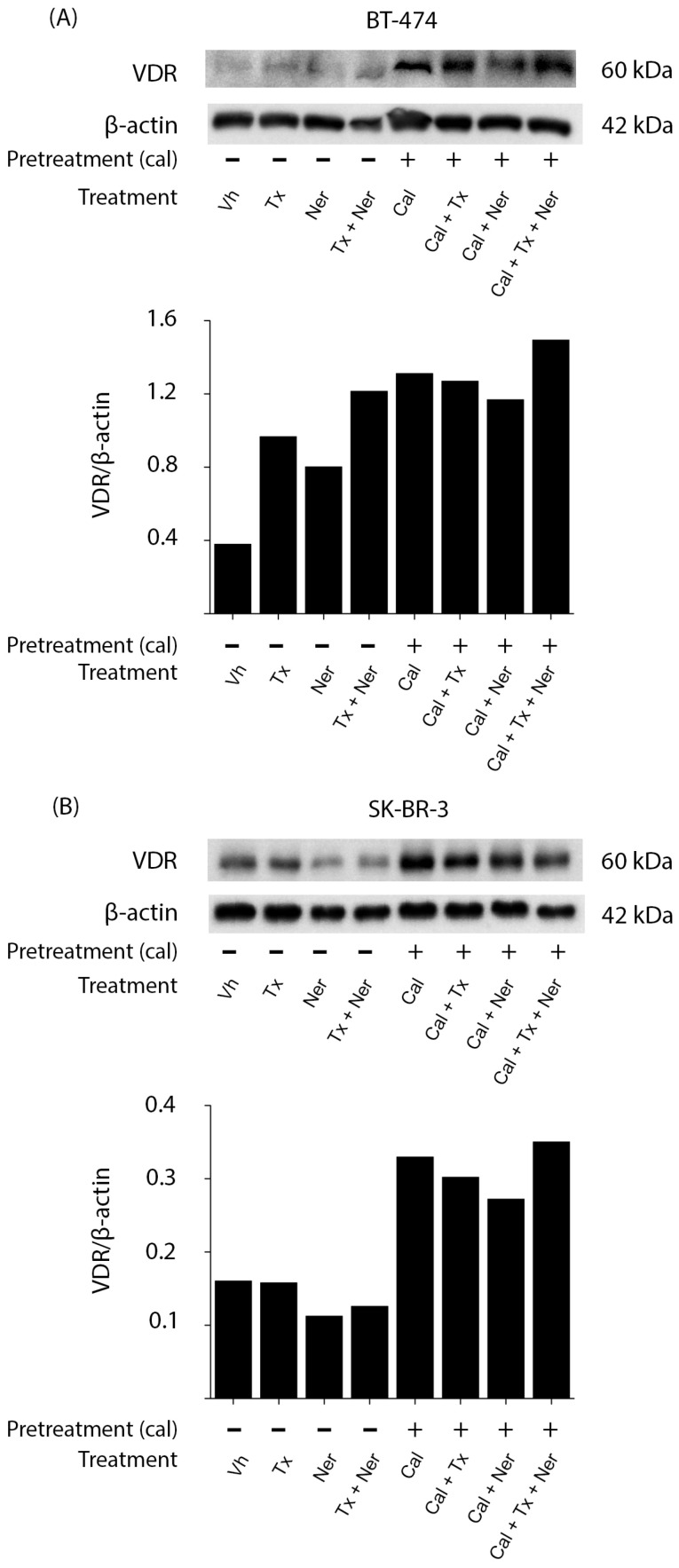
Effect of treatments on vitamin D receptor expression in HER2-positive breast cancer cell lines: (**A**) BT-474 and (**B**) SK-BR-3 cells were pretreated for 48 h in the absence (−) or presence of calcitriol (Cal (+); IC_50_ for each cell line). Cells were subsequently treated with tamoxifen (Tx; 1 × 10^−6^ M), neratinib (Ner; IC_50_ for each cell line), or their combination (Tx + Ner) in the absence or presence of calcitriol. Vitamin D receptor (VDR) protein levels were determined by Western blot, with β-actin used as a loading control. The figure shows a representative result from two independent experiments. Bar graphs represent the mean of the two replicates.

**Figure 4 ijms-26-08396-f004:**
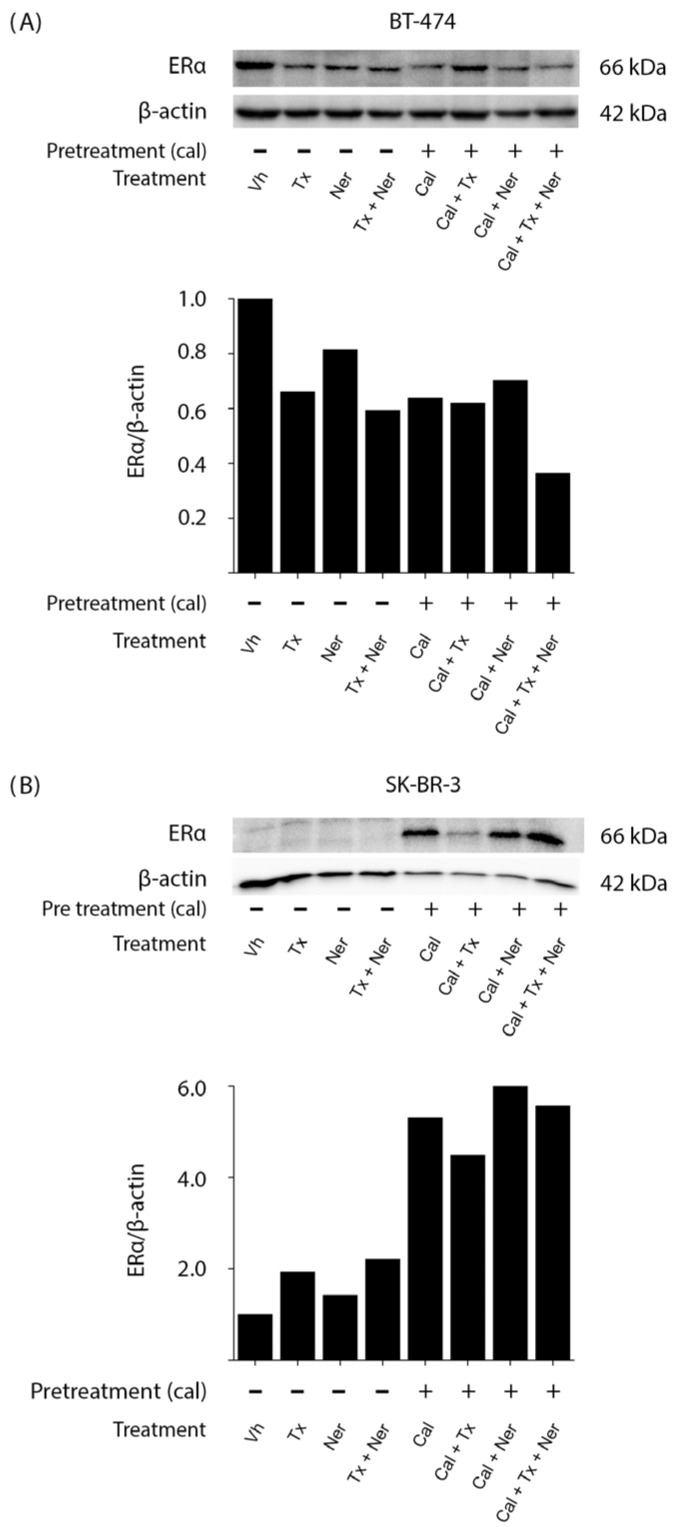
Modulation of ERα expression by calcitriol, tamoxifen, and neratinib in HER2-positive breast cancer cell lines: (**A**) BT-474 and (**B**) SK-BR-3 cells were pretreated for 48 h with vehicle (Vh, −) or calcitriol (Cal, +; IC_50_ as shown in Table 1). Cells were subsequently treated with tamoxifen (Tx; 1 × 10^−6^ M), neratinib (Ner; IC_50_), or their combination in the absence or presence of calcitriol. ERα protein levels were determined by Western blot, with β-actin used as a loading control. The figure shows a representative result from two independent experiments. Bar graphs represent the mean of the two replicates.

**Figure 5 ijms-26-08396-f005:**
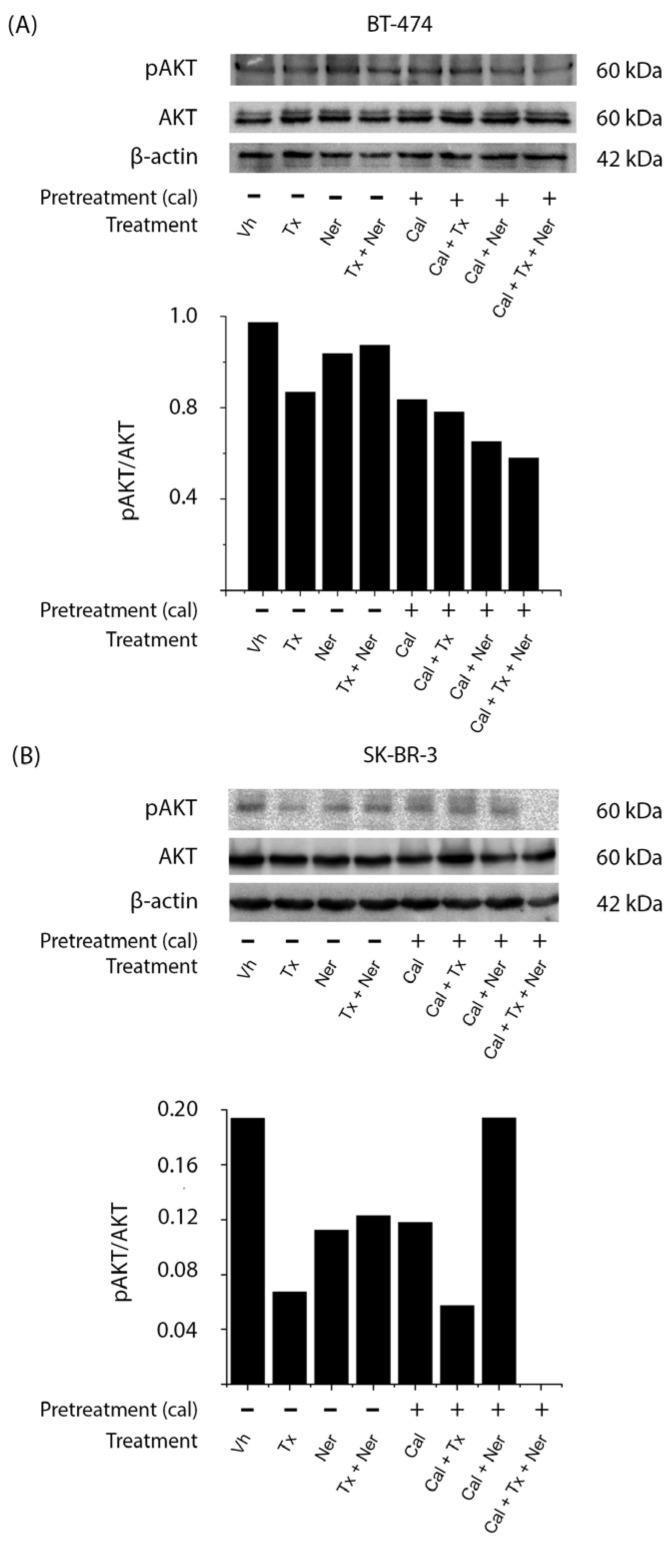
Effect of calcitriol in combination with tamoxifen and neratinib on AKT phosphorylation in HER2-positive breast cancer cells: (**A**) BT-474 and (**B**) SK-BR-3 cells were pretreated for 48 h with vehicle (Vh, −) or calcitriol (Cal, +; IC_50_ for each cell line) and subsequently treated with tamoxifen (Tx; 1 × 10^−6^ M), neratinib (Ner; IC_50_ for each cell line), or their combination in the absence or presence of calcitriol. Phosphorylated AKT (pAKT) protein levels were determined by Western blot, with β-actin used as a loading control. The figure shows a representative result from two independent experiments. Bar graphs represent the mean of the two replicates.

**Figure 6 ijms-26-08396-f006:**
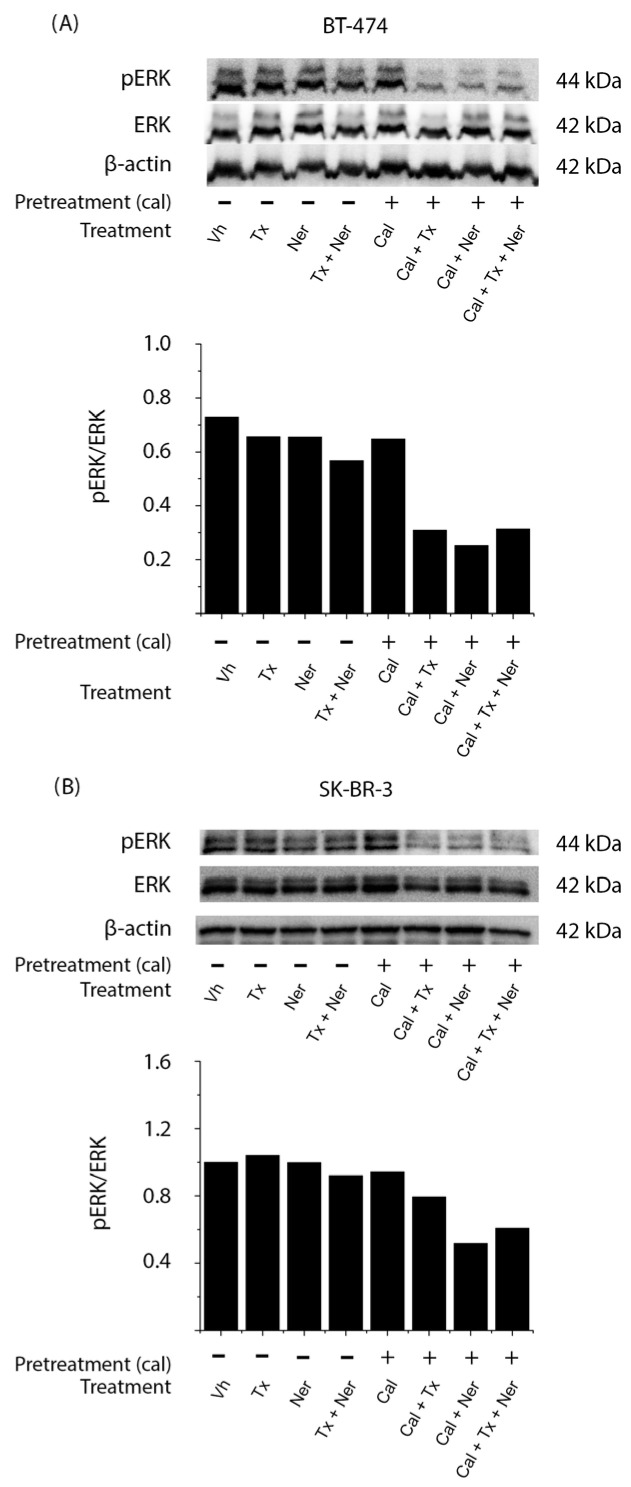
Effect of calcitriol in combination with tamoxifen and neratinib on ERK phosphorylation in HER2-positive breast cancer cells: (**A**) BT-474 and (**B**) SK-BR-3 cells were pretreated for 48 h with vehicle (Vh, −) or calcitriol (Cal, +; IC_50_ for each cell line). Cells were subsequently treated with tamoxifen (Tx; 1 × 10^−6^ M), neratinib (Ner; IC_50_ for each cell line), or their combination in the absence or presence of calcitriol. Phosphorylated ERK (pERK) protein levels were determined by Western blot, with β-actin used as a loading control. The figure shows a representative result from two independent experiments. Bar graphs represent the mean of the two replicates.

**Figure 7 ijms-26-08396-f007:**
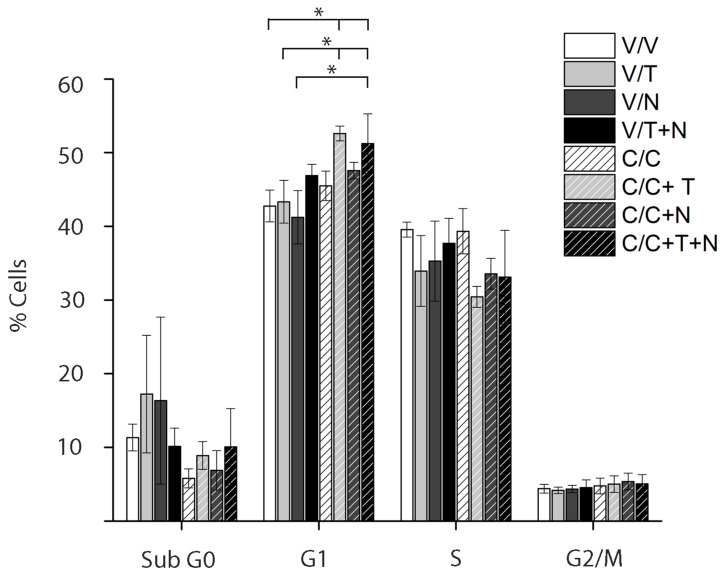
Effects of calcitriol, tamoxifen, and neratinib on cell cycle distribution in HER2-positive breast cancer cells. BT-474 cells were pretreated with vehicle (V) or calcitriol (C; IC_50_) for 48 h and subsequently treated with vehicle (V; white bar), tamoxifen (T; light gray bar), neratinib (N; IC_50_; dark gray bar), or their combination (T + N; black bar). Additional groups received calcitriol (white bar with stripes) in combination with each treatment (C + T, C + N, and C + T + N; corresponding bars with stripes). Cell cycle distribution was assessed by flow cytometry. Results are expressed as mean ± SEM of three independent experiments. * *p* ≤ 0.001.

**Table 1 ijms-26-08396-t001:** Inhibitory concentrations (IC)_20_ and IC_50_ values of calcitriol and neratinib in HER2-overexpressing breast cancer cell lines.

Compound	IC	BT-474 (mol/L)	SK-BR-3 (mol/L)
Calcitriol	20	8.7 × 10^−11^	8.4 × 10^−11^
	50	3.9 × 10^−9^	6.5 × 10^−9^
Neratinib	20	8.7 × 10^−10^	2.3 × 10^−10^
	50	3.8 × 10^−10^	4.8 × 10^−9^

## Data Availability

The authors confirm that the data supporting the findings of this study are available within the article.

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
