# Peer review of "Increased Antiproliferative Activity of Antiestrogens and Neratinib Treatment by Calcitriol in HER2-Positive Breast Cancer Cells"

_ijms, 2025, doi:10.3390/ijms26178396_

Round 1
Reviewer 1 Report
Comments and Suggestions for Authors
The paper entitled ‘Increased Antiproliferative Activity of Antiestrogens and 2 Neratinib Treatment by Calcitriol in HER2-Positive Breast Cancer Cells’ is very interesting as it provides evidence regarding the use of calcitriol as a promising adjuvant agent to enhance the efficacy of current therapeutic regimens in HER2-positive breast cancer. I believe that some modifications are necessary to improve the quality of the work.
1)- I consider that Lines 124-126, 132-137, 195-201, 218-222, 246-249, 258-262, 264-267, 268-274, 284-291, 302-307, 317-322, 331-337, 348-352, 366-368 should be removed from the results and included as part of the discussion.
2)- It would be necessary to clarify the results corresponding to Figure 2, as it is complex to understand which comparisons are and which are statistically different. Moreover, it would be important to unify Figures 2A and 2B as the comparisons are shown differently.
3)- It would be necessary to mention why they initially evaluate the effect of fulvestrant (Figure 2) and then in the rest of the experiments they only evaluate the effect of tamoxifen.
4)- Figures 3, 4, 5, 6 and 7 should specify which parameter the bars show. The description of the results should elaborate on the comparisons between the different groups. It is not mentioned or shown in the figures whether the differences are statistically significant, which complicates the understanding of the results and the scope of the results. I think it would be appropriate to include additional information on the variability or dispersion of the data presented in the figures mentioned (e.g. standard deviation, standard error or confidence intervals), as visualising the dispersion of the data would be very useful in assessing the robustness and reproducibility of the results.
5)- In lines 267-268 it mentions ‘Additionally, treatment with calcitriol (Cal) alone induced ERα expression, corroborating previous findings by Santos-Martínez et’. This is not clear in figure 4B.
6)- In lines 323-325 it mentions that ‘In BT-474 cells (Figure 6A), treatments in the absence of calcitriol, including tamoxifen (Tx), neratinib (Ner), and their combination (Tx + Ner), as well as calcitriol (Cal) alone, resulted in a moderate reduction of pERK levels compared to vehicle-treated controls’. These results are not visible in the graph as the differences are very small, and the dispersion of the results is not shown.
7)- In the statistical analysis, it would be important to specify whether, prior to the application of an ANOVA test, any test is carried out to assess whether the data are normally distributed or not.
8)- More than half of the bibliographical references are older than 10 years.
9)- Please note that Neratinib is not currently a first-line treatment for HER2+ breast cancer (https://doi.org/10.1016/j.breast.2024.103756).
Author Response
Reviewer 1
We thank Reviewer for the constructive feedback. We have addressed each of the comments point by point below.
The paper entitled ‘Increased Antiproliferative Activity of Antiestrogens and 2 Neratinib Treatment by Calcitriol in HER2-Positive Breast Cancer Cells’ is very interesting as it provides evidence regarding the use of calcitriol as a promising adjuvant agent to enhance the efficacy of current therapeutic regimens in HER2-positive breast cancer. I believe that some modifications are necessary to improve the quality of the work.
- I consider that Lines 124-126, 132-137, 195-201, 218-222, 246-249, 258-262, 264-267, 268-274, 284-291, 302-307, 317-322, 331-337, 348-352, 366-368 should be removed from the results and included as part of the discussion.
We thank the reviewer for this observation. In response to this comment, we carefully reviewed the indicated lines and agree that the interpretative and explanatory content in those sections is more appropriate for the discussion. Therefore, we have revised the manuscript accordingly and relocated the suggested sentences from the Results section to the Discussion section to maintain a more apparent distinction between the presentation of experimental findings and their interpretation. We believe this adjustment improves the structure and clarity of the manuscript. All changes have been highlighted in the revised version.
In the discussion section, the following was added:
Our study demonstrates that both calcitriol and neratinib exert concentration-dependent antiproliferative effects in HER2-positive breast cancer cell lines. Calcitriol exhibited similar IC₅₀ values in BT-474 and SK-BR-3 cells, indicating comparable sensitivity, while BT-474 cells were approximately ten times more sensitive to neratinib than SK-BR-3 cells. These differences may be attributed to intrinsic molecular characteristics of each cell line, such as ER expression, HER2 status, and tumor aggressiveness, which influence therapeutic response.
In ER-positive/HER2-positive breast cancer cells, the co-expression of ER and HER2 activates a complex network of signaling pathways characterized by bidirectional crosstalk, ultimately promoting tumor aggressiveness and therapeutic resistance. Consequently, recent therapeutic strategies have focused on the use of combination therapies that target multiple oncogenic pathways simultaneously to enhance efficacy. Such combination approaches aim to inhibit tumor growth and metastasis, arrest cell proliferation, reduce cancer stem cell populations, and induce apoptosis [42].
Specifically, in BT-474 cells, which co-express ERα and HER2, treatment with estradiol alone or in combination with neratinib significantly increased cell proliferation, underscoring the proliferative role of ER signaling in this subtype. In contrast, fulvestrant either alone or combined with neratinib markedly reduced proliferation, while tamoxifen had no significant effect, consistent with its partial agonist activity in certain cellular contexts [6, 43]. Calcitriol, administered alone or in combination with neratinib, significantly inhibited proliferation, and its addition to endocrine therapies further enhanced the antiproliferative response, regardless of the presence of estradiol. Notably, the triple combination of calcitriol, neratinib, and fulvestrant produced the strongest inhibitory effect on cell growth, suggesting a synergistic effect that counteracts estradiol-induced proliferation.
Importantly, these findings demonstrate that the antiproliferative effects of calcitriol can be maintained even in the presence of estradiol and may even enhance the antineoplastic effect of tamoxifen, either alone or in combination with estradiol. These results are consistent with previous studies, which have shown the importance of calcitriol in enhancing the inhibitory effects of tamoxifen in both ER-negative and HER2-positive breast cancer cells [27, 28].
In SK-BR-3 cells, which lack ERα expression, treatment with estradiol, antiestrogens, or their combinations did not significantly affect cell proliferation, as expected. In contrast, calcitriol and neratinib, either alone or in combination, effectively inhibited cell growth. The strongest antiproliferative effect was observed with the triple combination of calcitriol, neratinib, and fulvestrant. These results highlight the therapeutic potential of this strategy even in ER-negative/HER2-positive contexts and underscore the need for future in vivo studies in HER2-positive breast cancer models with differential ER expression. Such studies will be essential to evaluate the efficacy and translational relevance of triple therapies involving calcitriol, neratinib, and antiestrogens, particularly fulvestrant, in clinically relevant settings.
The re-sensitization to endocrine therapy and the enhanced antiproliferative response observed in SK-BR-3 cells may be attributed to the ability of calcitriol to restore ERα expression, as previously demonstrated in our laboratory [26]. This sensitizing effect is likely VDR-dependent, since in a previous study we showed that calcitriol increased ERα mRNA in a dose-dependent manner, an effect abolished by the VDR antagonist TEI-9647 [26]. Additionally, calcitriol exerts intrinsic antiproliferative effects that improve responses to conventional therapies used in breast cancer, such as neratinib and antiestrogens. In this context, several studies have shown that calcitriol enhances the antiproliferative efficacy of various antineoplastic agents when used in combination [19, 20, 28].
Collectively, these findings support the use of calcitriol as an adjuvant to improve the therapeutic efficacy of neratinib and antiestrogens in both ERα-positive and ERα-negative HER2-positive breast cancer.
Due to its widespread clinical use and robust evidence base, including randomized trials demonstrating that five years of tamoxifen treatment reduces breast cancer recurrence by approximately 40% and mortality by one-third, tamoxifen remains a cornerstone of adjuvant endocrine therapy in hormone receptor–positive breast cancer, particularly among premenopausal women [29, 30]. Therefore, we focused on tamoxifen in our mechanistic studies to evaluate its effects in combination with calcitriol and neratinib on the expression of key proteins involved in proliferation, invasion, and metastasis. This approach strengthens the translational relevance of our findings by aligning them with current therapeutic strategies.
Calcitriol exerts its biological effects primarily through binding to VDR, a nuclear transcription factor expressed in various tissues, including breast tissue. Upon activation, VDR regulates multiple cellular processes, including calcium transport, signal transduction, cell proliferation, and differentiation [44, 45]. In addition, VDR expression levels may correlate with the cells' responsiveness to calcitriol. In this study, we demonstrate that calcitriol upregulates VDR expression in HER2-positive breast cancer cells, regardless of their ER status.
In our study, the addition of calcitriol to tamoxifen and neratinib downregulated ERα protein levels in BT-474 cells (ER-positive/HER2-positive) and induced its expression in SK-BR-3 cells (ER-negative/HER2-positive). These results are consistent with previous reports showing that calcitriol modulates ERα expression depending on the initial receptor status [26, 27, 52]. In luminal cells, such as BT-474, the downregulation of ERα may be partially mediated by the binding of the calcitriol–VDR-RXR complex to negative vitamin D response elements (nVDREs) within the ESR1 promoter, leading to transcriptional repression [53].
Conversely, in SK-BR-3 cells, calcitriol alone or in combination induces ERα expression, corroborating previous findings by Santos-Martínez et al. [26], who demonstrated re-expression of ERα in ER-negative breast cancer cells following calcitriol treatment. This effect has been attributed to both direct transcriptional regulation and epigenetic modifications [27]. Notably, treatment with tamoxifen and its combination with neratinib also promoted ERα expression in SK-BR-3 cells, which may result from the inhibition of HER2-driven signaling and the activation of compensatory ERα pathways, highlighting the complex crosstalk between these signaling axes [54]. These results are consistent with previous observations from our laboratory, where we found that treatment with antiestrogens and the tyrosine kinase inhibitor lapatinib led to the upregulation of ERα in ER-negative/HER2-positive breast cancer cells [28].
Together, these findings indicate that calcitriol modulates ERα expression in a receptor status-dependent manner, potentially reversing features associated with endocrine resistance and aggressive tumor behavior. This context-dependent modulation underscores the potential of calcitriol as a therapeutic adjuvant to improve the efficacy of combined treatment strategies in HER2-positive breast cancer.
In breast cancer, particularly in the HER2-positive subtype, aberrant activation of the PI3K/AKT/mTOR and mitogen-activated protein kinase (MAPK/ERK) signaling pathways is frequently observed, promoting tumor cell proliferation, growth, migration, invasion, survival, and therapeutic resistance [55-60]. The PI3K/AKT/mTOR pathway regulates critical cellular processes, including growth, metabolism, and the inhibition of apoptosis, with AKT phosphorylation (pAKT) serving as a key marker of its activation. Similarly, the MAPK/ERK pathway, a highly interconnected cascade involving various kinases, plays a central role in oncogenesis and the development of drug resistance. Due to their relevance in breast cancer progression, these pathways have become major targets for therapeutic intervention [55-61]. In this context, we evaluated the effect of combining calcitriol with neratinib and tamoxifen on the phosphorylation of AKT and ERK in HER2-positive breast cancer cells. Our results demonstrate that treatment with calcitriol, either alone or in combination with tamoxifen and neratinib, reduced phosphorylation levels of both AKT and ERK in HER2-positive breast cancer cells. This suggests that calcitriol can disrupt key survival and proliferative signaling mechanisms, potentially enhancing the efficacy of antiestrogens and tyrosine kinase inhibitors.
These findings are consistent with previous reports by Segovia-Mendoza et al. [20], who showed that the combination of calcitriol and neratinib decreased pAKT and pERK levels in triple-negative breast cancer cells. Taken together, our results support the notion that calcitriol suppresses both PI3K/AKT and MAPK/ERK pathways in HER2-positive breast cancer cells, thereby reinforcing its potential as an adjuvant agent in combinatorial therapeutic strategies aimed at improving treatment outcomes and overcoming resistance. Considering these results, future studies evaluating the effects of calcitriol-based combinations on cell migration and invasion will be of great importance to further elucidate its role in limiting breast cancer progression.
These results suggest that the combination of calcitriol with tamoxifen and neratinib enhances the antiproliferative response primarily by promoting G1 cell cycle arrest and reducing S-phase progression in BT-474 cells. Notably, the triple treatment exerted the most pronounced effect, indicating that calcitriol enhances the efficacy of the combination therapy by modulating cell cycle dynamics rather than inducing cell death.
- It would be necessary to clarify the results corresponding to Figure 2, as it is complex to understand which comparisons are and which are statistically different. Moreover, it would be important to unify Figures 2A and 2B as the comparisons are shown differently.
Thank you for this valuable observation. We agree that the original presentation of Figure 2 may have complicated the interpretation of comparisons and statistical significance. To address this, we have revised and unified Figures 2A and 2B, presenting the results consistently.
Statistical analyses were performed as follows:
- Vehicle (V) vs. all treatments.
- Neratinib (N) vs. groups containing neratinib.
- Calcitriol (C) vs. groups containing calcitriol.
- Calcitriol + Neratinib (C+N) vs. groups containing this combination.
- For BT-474 cells: Estradiol (E) vs. all groups with estradiol, and Estradiol + Neratinib (E+N) vs. groups with estradiol.
- Within-group comparisons between treatments.
These details have been clarified in Figure 2 legend of the revised manuscript:
Statistical comparisons were performed as follows: V vs. all treatments; N, C, and C + N vs. groups containing N, C, and C + N, respectively; and in BT-474 cells, E and E+N vs. groups containing E and E+N, respectively. Within-group comparisons between treatments were also performed.
- It would be necessary to mention why they initially evaluate the effect of fulvestrant (Figure 2) and then in the rest of the experiments they only evaluate the effect of tamoxifen.
We appreciate the reviewer’s thoughtful observation. Given tamoxifen’s widespread clinical use and well-established efficacy, particularly in premenopausal patients with hormone receptor-positive breast cancer (https://doi.org/10.1200/OP.21.00384; https://doi.org/10.1016/S0140-6736(11)60993-8), we prioritized this antiestrogen in our initial mechanistic analyses. This allowed us to investigate the combined effects of calcitriol and neratinib on key proteins involved in proliferation, invasion, and metastasis under clinically relevant therapeutic conditions.
However, given the potent antiproliferative effect observed with the triple combination of calcitriol, neratinib, and fulvestrant, particularly in ERα-negative SK-BR-3 cells, we have extended our in vitro studies and initiated in vivo evaluations of this regimen. The enhanced response highlights its therapeutic potential even in ER-negative/HER2-positive contexts. These findings underscore the importance of conducting in vivo studies using HER2-positive breast cancer models with varying ER status to assess the efficacy and translational relevance of triple therapies involving calcitriol, neratinib, and antiestrogens, especially fulvestrant.
We have clarified this rationale in the revised manuscript, in the Methods, Results, and Discussion sections.
In the Methods section:
To assess the mechanistic effects on protein expression, the cells were pretreated with calcitriol and subsequently treated with tamoxifen, neratinib, or their combinations.
In the Results section:
Given tamoxifen’s widespread clinical use and well-established efficacy [29, 30], we focused the subsequent mechanistic analyses on this antiestrogen to investigate its combined effects with calcitriol and neratinib on key proteins involved in proliferation, invasion, and metastasis.
In the Discussion section:
In SK-BR-3 cells, which lack ERα expression, treatment with estradiol, antiestrogens, or their combinations did not significantly affect cell proliferation, as expected. In contrast, calcitriol and neratinib, either alone or in combination, effectively inhibited cell growth. The strongest antiproliferative effect was observed with the triple combination of calcitriol, neratinib, and fulvestrant. These results highlight the therapeutic potential of this strategy even in ER-negative/HER2-positive contexts and underscore the need for future in vivo studies in HER2-positive breast cancer models with differential ER expression. Such studies will be essential to evaluate the efficacy and translational relevance of triple therapies involving calcitriol, neratinib, and antiestrogens, particularly fulvestrant, in clinically relevant settings.
Due to its widespread clinical use and robust evidence base, including randomized trials demonstrating that five years of tamoxifen treatment reduces breast cancer recurrence by approximately 40% and mortality by one-third, tamoxifen remains a cornerstone of adjuvant endocrine therapy in hormone receptor–positive breast cancer, particularly among premenopausal women [29, 30]. Therefore, we focused on tamoxifen in our mechanistic studies to evaluate its effects in combination with calcitriol and neratinib on the expression of key proteins involved in proliferation, invasion, and metastasis. This approach strengthens the translational relevance of our findings by aligning them with current therapeutic strategies.
- Figures 3, 4, 5, 6 and 7 should specify which parameter the bars show. The description of the results should elaborate on the comparisons between the different groups. It is not mentioned or shown in the figures whether the differences are statistically significant, which complicates the understanding of the results and the scope of the results. I think it would be appropriate to include additional information on the variability or dispersion of the data presented in the figures mentioned (e.g. standard deviation, standard error or confidence intervals), as visualising the dispersion of the data would be very useful in assessing the robustness and reproducibility of the results.
We thank the reviewer for this observation. The number of biological replicates was specified in the figure legends as follows: in Figures 3 - 6, the legend indicates that “The figure shows a representative result from two independent experiments. Bar graphs represent the mean of the two replicates”. The bars in the graph represent the average of the two replicates.” For Figure 7, the legend states that “Results are expressed as mean ± SE of three independent experiments.” We will ensure that this information remains clear and consistent across all figure legends in the revised manuscript.
We agree that indicating data variability is important for assessing robustness. Although densitometric analysis was performed on two replicates, which limits formal statistical analysis, we have further elaborated on the comparative trends observed between the different treatment groups in the Results section. Where relevant, we also specify the magnitude and direction of changes in protein expression.
In Figures 3-6 was indicated: “The figure shows a representative result from two independent experiments. Bar graphs represent the mean of the two replicates”.
Figure 7 was indicated: “Results are expressed as mean ± SE of three independent experiments”.
- In lines 267-268 it mentions ‘Additionally, treatment with calcitriol (Cal) alone induced ERα expression, corroborating previous findings by Santos-Martínez et’. This is not clear in figure 4B.
Thank you for your observation. We agree that in the original version of Figure 4B the induction of ERα expression by calcitriol alone was not clearly visible in the blot. To improve data interpretation and better illustrate this result, we have replaced the blot with a more representative image from an independent replicate, where the increase in ERα levels following calcitriol treatment is more evident. This change allows for clearer visualization of the effect described in lines 267–268 and aligns with the findings previously reported by Santos-Martínez et al.
New figure:
- In lines 323-325 it mentions that ‘In BT-474 cells (Figure 6A), treatments in the absence of calcitriol, including tamoxifen (Tx), neratinib (Ner), and their combination (Tx + Ner), as well as calcitriol (Cal) alone, resulted in a moderate reduction of pERK levels compared to vehicle-treated controls’. These results are not visible in the graph as the differences are very small, and the dispersion of the results is not shown.
Thank you for your comment. We agree that the differences in ERK phosphorylation levels among the groups treated with tamoxifen (Tx), neratinib (Ner), and their combination (Tx + Ner) in the absence of calcitriol were minimal and not clearly appreciable in the graph. In light of this, we have revised the manuscript and modified the original statement, which previously referred to a “moderate reduction”
In BT-474 cells (Figure 6A), treatments in the absence of calcitriol, including tamoxifen (Tx), neratinib (Ner), and their combination (Tx + Ner), as well as calcitriol (Cal) alone, did not modify pERK levels compared to vehicle-treated controls. In contrast, a pronounced decrease in ERK phosphorylation was observed when calcitriol was administered in combination with either tamoxifen (Cal + Tx), neratinib (Cal + Ner), or both (Cal + Tx + Ner), as confirmed by densitometric analysis normalized to total ERK.
- In the statistical analysis, it would be important to specify whether, prior to the application of an ANOVA test, any test is carried out to assess whether the data are normally distributed or not.
We thank the reviewer for highlighting the importance of verifying statistical assumptions. Prior to applying ANOVA, we assessed the normality of each dataset using the Kolmogorov-Smirnov test. The results showed that data from the SKBR3 cell line groups conformed to a normal distribution, while those from the BT-474 cell line groups did not. Accordingly, we applied parametric ANOVA to the SK-derived data and non-parametric Kruskal-Wallis One-Way Analysis of Variance on Ranks tests to the BT-derived data.
This methodological distinction has now been explicitly stated in the revised Methods section:
“Prior to applying ANOVA, we assessed the normality of each dataset using the Kolmogorov-Smirnov test. The results showed that data from the SK-BR-3 cell line groups conformed to a normal distribution, while those from the BT-474 cell line groups did not. Accordingly, we applied parametric ANOVA to the SK-BR-3 derived data and non-parametric Kruskal-Wallis One Way Analysis of Variance on Ranks tests to the BT-474 derived data.”
- More than half of the bibliographical references are older than 10 years.
Thank you for emphasizing the importance of updating our bibliography. We have carefully revised the reference list, incorporating recent high-quality studies where appropriate, while preserving essential foundational works.
These additions complement our classical references, ensuring that our manuscript reflects both the historical context and the most recent evidence in the field.
New references:
- Swain, S.M., M. Shastry, and E. Hamilton, Targeting HER2-positive breast cancer: advances and future directions. Nature Reviews Drug Discovery, 2023. 22(2): p. 101-126.
- Singh, H., et al., U.S. Food and Drug Administration Approval: Neratinib for the Extended Adjuvant Treatment of Early-Stage HER2-Positive Breast Cancer. Clin Cancer Res, 2018. 24(15): p. 3486-3491.
- Administration, U.S.F.D. FDA approves neratinib for metastatic HER2-positive breast cancer. 2020 [cited 2025 July 20]; Available from: https://www.fda.gov/drugs/resources-information-approved-drugs/fda-approves-neratinib-metastatic-her2-positive-breast-cancer.
- Agency, E.M. Nerlynx: Summary of product characteristics. 2020 [cited 2025 July 21]; Available from: https://www.ema.europa.eu/en/medicines/human/EPAR/nerlynx.
- Pegram, M., C. Jackisch, and S.R.D. Johnston, Estrogen/HER2 receptor crosstalk in breast cancer: combination therapies to improve outcomes for patients with hormone receptor-positive/HER2-positive breast cancer. NPJ Breast Cancer, 2023. 9(1): p. 45.
- Negri, M., et al., Vitamin D-Induced Molecular Mechanisms to Potentiate Cancer Therapy and to Reverse Drug-Resistance in Cancer Cells. 2020. 12(6): p. 1798.
- Carlberg, C., Vitamin D and Its Target Genes. Nutrients, 2022. 14(7).
- McAndrew, N.P. and R.S. Finn, Clinical Review on the Management of Hormone Receptor–Positive Metastatic Breast Cancer. 2022. 18(5): p. 319-327.
- Early Breast Cancer Trialists' Collaborative, G., Relevance of breast cancer hormone receptors and other factors to the efficacy of adjuvant tamoxifen: patient-level meta-analysis of randomised trials. The Lancet, 2011. 378(9793): p. 771-784.
- Emde, A., et al., Simultaneous Inhibition of Estrogen Receptor and the HER2 Pathway in Breast Cancer: Effects of HER2 Abundance. Transl Oncol, 2011. 4(5): p. 293-300.
- Kim, N. and K.E. Lukong, Treating ER-positive breast cancer: a review of the current FDA-approved SERMs and SERDs and their mechanisms of action. 2025. Volume 19 - 2025.
- Braicu, C., et al., A Comprehensive Review on MAPK: A Promising Therapeutic Target in Cancer. Cancers (Basel), 2019. 11(10).
- Shin, Y., et al., TMBIM6-mediated miR-181a expression regulates breast cancer cell migration and invasion via the MAPK/ERK signaling pathway. J Cancer, 2023. 14(4): p. 554-572.
- Xu, J., et al., How is the AKT/mTOR pathway involved in cell migration and invasion? Biocell, 2023. 47(4): p. 773.
- Please note that Neratinib is not currently a first-line treatment for HER2+ breast cancer (https://doi.org/10.1016/j.breast.2024.103756).
Thank you for your observation. We agree that neratinib is not currently indicated as a first-line treatment for HER2-positive breast cancer. In response to this comment, we have revised the Introduction to accurately reflect its approved clinical indications. Specifically, we incorporated the following information:
“Neratinib has been approved by the U.S. FDA for extended adjuvant treatment following trastuzumab-based therapy in early-stage HER2-positive breast cancer, and in combination with capecitabine for patients with advanced or metastatic HER2-positive disease who have received at least two prior anti-HER2 therapies [13, 14]. Similarly, the European Medicines Agency approved its use in hormone receptor–positive, HER2-positive early breast cancer as extended adjuvant therapy [15].”

Reviewer 2 Report
Comments and Suggestions for Authors
In the manuscript “Increased Antiproliferative Activity of Antiestrogens and Neratinib Treatment by Calcitriol in HER2-Positive Breast Cancer Cells”, Edgar Milo-Rocha et al., have discovered the potential of Calcitriol, the active form of vitamin D, induced ERα expression in ER-negative breast cancer cells, thereby sensitizing them to the antiproliferative effects of antiestrogens. When combined with the tyrosine kinase inhibitor anti-cancer medication - neratinib, calcitriol enhanced cell growth inhibition. Therefore, authors investigated whether adding calcitriol to the combined treatment with antiestrogens and neratinib could further inhibit the proliferation of HER2-positive breast cancer cells, regardless of their ER status.
The authors used BT-474 (ER-positive/HER2-positive) and SK-BR-3 (ER-negative/HER2-positive) breast cancer cell lines and treated with calcitriol to modulate ER expression. Following the treatment with calcitriol in combination with neratinib, with or without antiestrogens. Using the functional assays like proliferation, cell cycle analysis, and Western blotting demonstrated that calcitriol and neratinib significantly inhibit cell proliferation in a concentration-dependent manner in the ER2-positive cell lines. Calcitriol enhanced the antiproliferative response of combined neratinib and antiestrogen treatment. Calcitriol, alone or in combination, modulated vitamin D receptor and ERα expression, reduced AKT and ERK phosphorylation, and promoted G1 phase arrest while decreasing the S phase population. These findings support the potential of this combinatorial approach as a therapeutic strategy for HER2- positive breast cancer.
Minor comments:
1) Is the effect of calcitriol reversible: Did the authors try to was the calcitriol pre-treated cells with fresh media before adding the antiestrogens and neratinib. This could strongly demonstrate if the effect of calcitriol is reversible or not?
2) In figure 3, 4, 5, 6 and 7: How many biological replicates were performed for the western blot experiments? This needs to be mentioned in the figure legends.
3) Have the authors performed the invasion of the BT-474 and SK-BR-3 cells with calcitriol to the combined treatment with antiestrogens and neratinib?
4) Does the author have any support of the in vivo data from mouse model to justify the discovery in the more relevant physiological setting atleast from the point of view of phenotype.
Author Response
Reviewer 2
We thank Reviewer for the constructive feedback. We have addressed each of the comments point by point below.
In the manuscript “Increased Antiproliferative Activity of Antiestrogens and Neratinib Treatment by Calcitriol in HER2-Positive Breast Cancer Cells”, Edgar Milo-Rocha et al., have discovered the potential of Calcitriol, the active form of vitamin D, induced ERα expression in ER-negative breast cancer cells, thereby sensitizing them to the antiproliferative effects of antiestrogens. When combined with the tyrosine kinase inhibitor anti-cancer medication - neratinib, calcitriol enhanced cell growth inhibition. Therefore, authors investigated whether adding calcitriol to the combined treatment with antiestrogens and neratinib could further inhibit the proliferation of HER2-positive breast cancer cells, regardless of their ER status.
The authors used BT-474 (ER-positive/HER2-positive) and SK-BR-3 (ER-negative/HER2-positive) breast cancer cell lines and treated with calcitriol to modulate ER expression. Following the treatment with calcitriol in combination with neratinib, with or without antiestrogens. Using the functional assays like proliferation, cell cycle analysis, and Western blotting demonstrated that calcitriol and neratinib significantly inhibit cell proliferation in a concentration-dependent manner in the ER2-positive cell lines. Calcitriol enhanced the antiproliferative response of combined neratinib and antiestrogen treatment. Calcitriol, alone or in combination, modulated vitamin D receptor and ERα expression, reduced AKT and ERK phosphorylation, and promoted G1 phase arrest while decreasing the S phase population. These findings support the potential of this combinatorial approach as a therapeutic strategy for HER2- positive breast cancer.
Minor comments:
- Is the effect of calcitriol reversible: Did the authors try to was the calcitriol pre-treated cells with fresh media before adding the antiestrogens and neratinib. This could strongly demonstrate if the effect of calcitriol is reversible or not?
We appreciate this insightful comment. We did not perform the experiment in which calcitriol-pretreated cells were washed with fresh media before the addition of antiestrogens and neratinib. However, we consider that the sensitizing effect of calcitriol to antiestrogens is likely VDR-dependent. In our previous study (https://doi.org/10.1186/1471-2407-14-230), we demonstrated that calcitriol significantly increased ERα mRNA in a dose-dependent manner. This effect was specifically mediated through the VDR, since the VDR antagonist TEI-9647 significantly abolished the stimulatory effect of calcitriol on ERα gene expression. Based on these findings, we infer that the effect of calcitriol on sensitizing cells to antiestrogens is mediated through VDR activity. We agree with the reviewer that testing the reversibility of the effect would provide valuable information, and we will incorporate this important consideration into the Discussion section of the revised manuscript.
In Discussion section: “The re-sensitization to endocrine therapy and the enhanced antiproliferative response observed in SK-BR-3 cells may be attributed to the ability of calcitriol to restore ERα expression, as previously demonstrated in our laboratory [26]. This sensitizing effect is likely VDR-dependent, since in a previous study we showed that calcitriol increased ERα mRNA in a dose-dependent manner, an effect abolished by the VDR antagonist TEI-9647 [26].”
- In figure 3, 4, 5, 6 and 7: How many biological replicates were performed for the western blot experiments? This needs to be mentioned in the figure legends.
We thank the reviewer for this observation. The number of biological replicates was specified in the figure legends as follows: in Figures 3 - 6, the legend indicates that “the figure shows a representative result from two independent experiments. The bars in the graph represent the average of the two replicates.” For Figure 7, the legend states that “results are expressed as mean ± SE of three independent experiments.” We will ensure that this information remains clear and consistent across all figure legends in the revised manuscript.
In Figures 3-6 was indicated: “The figure shows a representative result from two independent experiments. Bar graphs represent the mean of the two replicates”.
Figure 7 was indicated: “Results are expressed as mean ± SE of three independent experiments”.
- Have the authors performed the invasion of the BT-474 and SK-BR-3 cells with calcitriol to the combined treatment with antiestrogens and neratinib?
We appreciate the reviewer’s suggestion. We did not perform invasion assays in BT-474 or SK-BR-3 cells treated with calcitriol in combination with antiestrogens and neratinib. However, we agree that this is an important aspect to explore, and future studies will address whether calcitriol can also modulate the invasive potential of HER2-positive breast cancer cells under these treatment conditions.
In the discussion section it was added the following paragraph:
In breast cancer, particularly in the HER2-positive subtype, aberrant activation of the PI3K/AKT/mTOR and mitogen-activated protein kinase (MAPK/ERK) signaling pathways is frequently observed, promoting tumor cell proliferation, growth, migration, invasion, survival, and therapeutic resistance [55-60]. The PI3K/AKT/mTOR pathway regulates critical cellular processes, including growth, metabolism, and the inhibition of apoptosis, with AKT phosphorylation (pAKT) serving as a key marker of its activation. Similarly, the MAPK/ERK pathway, a highly interconnected cascade involving various kinases, plays a central role in oncogenesis and the development of drug resistance. Due to their relevance in breast cancer progression, these pathways have become major targets for therapeutic intervention [55-61]. In this context, we evaluated the effect of combining calcitriol with neratinib and tamoxifen on the phosphorylation of AKT and ERK in HER2-positive breast cancer cells. Our results demonstrate that treatment with calcitriol, either alone or in combination with tamoxifen and neratinib, reduced phosphorylation levels of both AKT and ERK in HER2-positive breast cancer cells. This suggests that calcitriol can disrupt key survival and proliferative signaling mechanisms, potentially enhancing the efficacy of antiestrogens and tyrosine kinase inhibitors.
These findings are consistent with previous reports by Segovia-Mendoza et al. [20], who showed that the combination of calcitriol and neratinib decreased pAKT and pERK levels in triple-negative breast cancer cells. Taken together, our results support the notion that calcitriol suppresses both PI3K/AKT and MAPK/ERK pathways in HER2-positive breast cancer cells, thereby reinforcing its potential as an adjuvant agent in combinatorial therapeutic strategies aimed at improving treatment outcomes and overcoming resistance. Considering these results, future studies evaluating the effects of calcitriol-based combinations on cell migration and invasion will be of great importance to further elucidate its role in limiting breast cancer progression.
- Does the author have any support of the in vivo data from mouse model to justify the discovery in the more relevant physiological setting atleast from the point of view of phenotype.
We appreciate the reviewer’s observation. At present, our study was limited to in vitro analyses in HER2-positive breast cancer cell lines, and no in vivo data were included. However, we fully agree that validation in physiologically relevant models is essential to confirm the translational applicability of our findings.Accordingly, we have now emphasized this point in the Discussion section by adding the following statement:
“These results highlight the therapeutic potential of this strategy even in ER-negative/HER2-positive contexts and underscore the need for future in vivo studies in HER2-positive breast cancer models with differential ER expression. Such studies will be essential to evaluate the efficacy and translational relevance of triple therapies involving calcitriol, neratinib, and antiestrogens, particularly fulvestrant, in clinically relevant settings.”

Round 2
Reviewer 1 Report
Comments and Suggestions for Authors
Thank you for taking my suggestions and proposals for improvement. Congratulations. Good luck with the publication of your work.